# Interventional Influence of the Intestinal Microbiome Through Dietary Intervention and Bowel Cleansing Might Improve Motor Symptoms in Parkinson’s Disease

**DOI:** 10.3390/cells9020376

**Published:** 2020-02-06

**Authors:** Tobias Hegelmaier, Marco Lebbing, Alexander Duscha, Laura Tomaske, Lars Tönges, Jacob Bak Holm, Henrik Bjørn Nielsen, Sören G. Gatermann, Horst Przuntek, Aiden Haghikia

**Affiliations:** 1Department of Neurology, Ruhr-University Bochum, St. Josef-Hospital Bochum, Bochum, 44791, Germany; tobias.hegelmaier@googlemail.com (T.H.); Alexander.Duscha@ruhr-uni-bochum.de (A.D.); Laura.Ackermann@ruhr-uni-bochum.de (L.T.); lars.toenges@rub.de (L.T.); 2Clinic of Neurology II, EVK Hattingen, Hattingen 45525, Germany; marco.lebbing@googlemail.com (M.L.); przuntekh@t-online.de (H.P.); 3Clinical Microbiomics A/S, Copenhagen 2200, Denmark; jacob@clinical-microbiomics.com (J.B.H.); hbjorn@clinical-microbiomics.com (H.B.N.); 4Department of Medical Microbiology, Ruhr-University Bochum, Bochum, 44801, Germany; soeren.gatermann@rub.de

**Keywords:** vegetarian diet, enema, Parkinson’s disease, microbiome, butyric acid

## Abstract

The impact of the gut microbiome is being increasingly appreciated in health and in various chronic diseases, among them neurodegenerative disorders such as Parkinson’s disease (PD). In the pathogenesis of PD, the role of the gut has been previously established. In conjunction with a better understanding of the intestinal microbiome, a link to the misfolding and spread of alpha-synuclein via inflammatory processes within the gut is discussed. In a case-control study, we assessed the gut microbiome of 54 PD patients and 32 healthy controls (HC). Additionally, we tested in this proof-of-concept study whether dietary intervention alone or additional physical colon cleaning may lead to changes of the gut microbiome in PD. 16 PD patients underwent a well-controlled balanced, ovo-lacto vegetarian diet intervention including short fatty acids for 14 days. 10 of those patients received additional treatment with daily fecal enema over 8 days. Stool samples were collected before and after 14 days of intervention. In comparison to HC, we could confirm previously reported PD associated microbiome changes. The UDPRS III significantly improved and the levodopa-equivalent daily dose decreased after vegetarian diet and fecal enema in a one-year follow-up. Additionally, we observed a significant association between the gut microbiome diversity and the UPDRS III and the abundance of *Ruminococcaceae*. Additionally, the abundance of *Clostridiaceae* was significantly reduced after enema. Dietary intervention and bowel cleansing may provide an additional non-pharmacologic therapeutic option for PD patients.

## 1. Introduction

Parkinson’s disease (PD) is one of the most common neurodegenerative disorders only second to Alzheimer’s disease and the most common movement disorder worldwide [1]. The cause of non-monogenic forms of PD is still not fully understood. PD is a multifactorial disorder with a strong environmental component. Mitochondrial dysfunction, oxidative stress, intracellular protein accumulation and abnormal protein degradation are the pathogenic hallmarks of PD [2]. Although the diagnosis of PD is mainly based on the clinical manifestation of extrapyramidal motor symptoms, non-motor symptoms often precede the diagnosis by decades [1]. These include deficits of the autonomic nervous system, olfactory function and symptoms linked to lower brainstem and cerebral cortex. Gastrointestinal dysfunction, for example, obstipation affects more than 80% of patients with PD [1,3].

There is growing evidence that the microbiome is altered in numerous neurodegenerative diseases, in particular Parkinson’s disease. In addition to an overall dysbiosis, the most consistent changes were, besides an increase in the relative abundances of *Verrucomicrobiaceae* and *Akkermansia*, a decrease of *Prevotellacae* [4,5,6,7,8,9]. In this context, Keshavarzian et al. found a pro-inflammatory microbiome with an increase of *Proteobacteria* of the genus *Ralstonia* while the potential anti-inflammatory butyrate-producing bacteria *Blautia*, *Coprococcus* and *Roseburia* were reduced in PD [7,8]. For a long time now, it has been hypothesized that the gut may play a crucial role in the etiology of PD. The intestinal microbiome with a similar amount of bacterial cells as human cells, could be the decisive factor linking the environment and neuronal damage in the central nervous system (CNS) of PD patients [10,11,12]. According to the first theory, alpha-synuclein accumulates in the enteric nervous system and retrogradely extends to the brain stem via the vagus nerve and its nucleus dorsalis [13,14]. Further support for the gut-theory can be deduced from the epidemiological observation that vagotomized individuals have a reduced risk of PD [15]. However, this hypothesis was challenged by a Swedish register-based matched-cohort study that could not detect a reduced risk for the development of PD by vagotomy in general. Nevertheless, a sub analysis revealed a reduced risk through a truncal vagotomy [16]. In recent years, alternative ways of communication between the intestine and the brain have been discussed. In addition to the neural path, there is evidence that the endocrine and blood system plays a decisive role [17]. According to the proinflammatory gut microbiome growing evidence support the hypothesis that inflammatory processes play an important role in PD and might be a further pathway of the gut-brain-axis [18,19]. Supporting this notion, different retrospective studies exposed a higher incidence of PD in patients with chronic inflammatory bowel diseases [20]. This hypothesis coincides with the detection of proinflammatory markers in stool samples. Calprotectin is a marker for increased activity of the immune system, which is used to monitor inflammatory bowel disease (IBD). It has been shown that calprotectin is also elevated in the stool samples of patients with PD [21,22]. Further inflammatory markers (Il-1b, Il-1a, fecal C-reactive protein and CXCL8 (Il-8)) have been detected by Houser et al. in stool and by Deros et al. in colon biopsies of PD patients [23,24]. In addition, increased permeability markers in the feces of PD patients were also detected with zonulin and alpha-trypsin [21]. A higher permeability and disrupted occludin, controlling intestinal barrier permeability, have been found in PD patients [25,26], which may lead to increased transition of pro-inflammatory cytokines that have been observed within the colon of patients with PD [24]. These results strongly support the hypothesis that micro inflammation with increased intestinal permeability and a potential leaky-gut syndrome with the absorption of endotoxins plays a crucial role in PD. Additional to the proinflammatory intestinal environment, there are indications that the increased inflammatory activity is associated with an elevated concentration of alpha-synuclein. According to this Keshavarzian et al. hypothesized that the dysbiosis could potentially trigger inflammation-induced misfolding of a-Syn in PD [8,27].

In addition to increased local inflammatory activity in the intestine, there is also evidence of increased systemic inflammatory activity in Parkinson’s disease. Thus, the working group around Williams-Grey and Bordacki detected increased levels of proinflammatory cytokines such as TNF-a, INF-gamma IL1-b, IL-2, IL-4, IL-6 and IL-10 within the blood of PD patients [28,29]. In this context, it has already been suspected that proinflammatory cytokines lead to a disruption of the blood-brain barrier and consequently are indirectly involved in the activation of glial cells [18,19].

Besides inflammatory changes within all levels of the gut-brain axis further evidence pointing to a major impact of the microbiome on the clinical course. Therefore, in a mouse model of PD transplantation of the microbiome of patients with PD into receptive mice led to motor symptoms as opposed to the transplant sample of healthy individuals [30]. Furthermore, recently it has also been shown that gut-associated bacteria differ in their ability to decarboxylate levodopa to dopamine via tyrosine decarboxylases in the proximal small intestine and thereby influence the levodopa dosages needed for symptom control [31]. In this regard, a conserved tyrosine decarboxylase (TyrDC) has been detected within Enterococcus faecalis [32]. In addition, a negative correlation between the frequency of *Firmicutes* and the use of entacapone has been reported [7]. These data suggest that in future the microbiome might be a crucial component in the management of side effects as well as in the therapy of PD.

The influence of the microbiome on the clinical course raises the question of potential therapies that can positively influence the microbiome. It has been researched that a vegetarian diet causes an anti-inflammatory effect in healthy subjects. With regard to the proinflammatory intestinal environment in PD, we investigated the influence of a vegetarian diet including a high proportion of anti-inflammatory acting short-fatty acids (SCFA) on the microbiome and the clinical course in patients with PD. Another therapy for IBD are rectal enemas. For this reason, in this proof-concept study, we have also looked in a small cohort at the influence of enemas on the microbiome and the clinical course. 

## 2. Materials and Methods

### 2.1. Participants

The study was performed from November 2013 to January 2015 (gut microbiome analysis was approved by the Ethics Committee of the Ruhr-University Bochum; registration number 4493-12). Patients were recruited from the Department of Neurology of the University Hospital St. Josef-Hospital Bochum and from the Clinic of Neurology in Hattingen (Klinik für Neurologie II und Komplementärmedizin, Evangelisches Krankenhaus Hattingen, Germany). Written informed consent was obtained from all patients and healthy participants. 

Inclusion criteria for PD patients comprised the diagnosis of non-monogenic forms of PD. Further, all patients were independently mobile and completely orientated for time, location, person and situation. Exclusion criteria comprised poor general condition, serious concomitant disease or organ dysfunction, insufficient German language skills, severe psychiatric diseases, serious pre-existing conditions of the gastrointestinal tract, malignancies, pregnancy and missing informed consent. 

Healthy subjects served as controls (HC) and were at an age between 18 to 85 years. HC were excluded when any of the following diseases were present: chronic gastrointestinal disease; chronic systemic autoimmune disease with GI involvement; neurological disease; psychiatric diseases; malignancies. Furthermore, the detection of systemic infections or pathogenic bacterial strains in feces or systemically, and holiday history in risk areas (tropical countries, countries with risk of travel diarrhea) within the last 6 months or medication history affecting the microbiome, that is, antibiotics within the last 6 months, were considered as exclusion criteria.

For microbiome analysis, we enrolled 54 patients with idiopathic PD and 32 HC (demographic details enlisted in Table 1). Additionally, from 16 of these patients a stool sample was taken before and after 14 days of dietary intervention with or without bowel cleansing (demographic details enlisted in Table 2). The detailed schedule is shown graphically in Figure 1.

### 2.2. Bowel Cleansing

In advance, all patients received safety examinations including electrocardiography, as well as routine blood tests. During the procedure the blood pressure, oxygen saturation and pulse rate was measured.

Since there is no international standard procedure for enema, we decided to use an oil enema for an adequate lubrication and as a stool softener because of the increased obstipation tendency in PD. Bowel cleansing was performed on 8 consecutive days. Before the procedure, the decoction was prepared with warm water with electrolytes (water temperature was adjusted to 38 °C). Furthermore, 120 mL oil was continuously stirred until a uniform consistency was attained. Nursing care, including psychologic care and abdominal massage was provided during the entire procedure.

Between 11.00 to 12.00 am, shortly before lunch, the procedure was performed according to the following standard operating procedures: Patients were placed in a supine position on a bed. A small amount of oil was applied to the anal canal for lubrication, then a rectal examination was carried out to rule out loaded rectum or any other obstruction. Afterwards the tip of the irrigator was gently introduced into the anus and the height was unadjusted slowly and steadily to allow the decoction into the patient’s rectum. Patients were instructed to breathe deeply to relax the anal canal before insertion. Until the urge of defecation occurs, the patients were asked to lie in supine position. The procedure lasted for approximately 30 min. 

Possible contraindications for enemas were multiple abdominal surgery in history, gastrointestinal bleeding, vomiting, abdominal pain, malignancies, as well as hemorrhoids and severe heart disease.

Possible side effects of the enemas are orthostatic dysregulation, electrolyte imbalance, exsiccosis, infections, abdominal pain, diarrhea, nausea and vomiting.

### 2.3. Dietary Intervention 

All patients received a balanced, vegetarian regimen in an ovo-lacto vegetarian style including ghee during the observation period of 14 days. The participants received three meals a day. One meal in the morning, noon and evening. 45% of the dishes contained ghee that contains up to 30% butyric acid. All ingredients including spices, herbs, oils, seeds, nuts and kernels of all meals during the 14 days are listed in Table 3.

### 2.4. Bristol-Stool-Scale

The Bristol-stool-scale is a diagnostical tool, designed by the Bristol Royal Infirmary, to classify the form of the human faces into seven categories. This well-established scale is used in clinical and experimental settings and was additionally assessed in PD and HC [33].

### 2.5. Sample Acquisition, Preparation and 16sRNA Sequencing

All patients’ fecal samples were collected at the Clinic of Neurology II, EVK Hattingen Germany and were snap frozen immediately at −80 °C after defecation. In 16 patients with PD, a sample was collected before, during and after combined treatment including dietary intervention and bowel cleansing. 

For the extraction of DNA, a commercial system (QIAamp Fast Stool Mini Kit, Qiagen, Hilden, Germany) was used. In brief, the samples were thawed at 4 °C and approximately 500 mg was suspended in Inhibitex buffer (3 mL). Suspensions were heated (95 °C, 5 min), spun (20,000× *g*) and the supernatant was used for DNA preparation according to the protocol of the manufacturer. Purity was checked by determining the OD260/OD280 ratio, damage and degradation by agarose gel electrophoresis and absence of inhibitors was verified by amplification with the primers 27F and 534R, which amplify the V1-V3 regions. The 16SrDNA region was amplified with the same primers and sequenced by Illumina technology by a commercial vendor (GATC Biotech, Konstanz, Germany).

### 2.6. Unified Parkinson Disease Rating Scale

The Unified Parkinson Disease Ratings Scale (UPDRS) is the most widely used scale for the clinical evaluation of PD patients. There is UPDRS I-IV. UPDRS I evaluate mentation, behavior and mood, II includes a self-assessment about daily life experience and IV about complications of therapy. UPDRS III is an internationally well-established rating scale for assessing the motor symptoms of patients with PD. It includes assessment of speech, facial expression, rest tremor (face, hands and feet), action or postural tremor of the hands, rigor (neck, arms and legs), finger dexterity (finger tapping), hand movements (opening and closing the hands), rapidly changing hand movements (pronation and supination movements), agility of the legs, getting up from the chair, posture, gait, postural stability and brady- and hypokinesia of the body. A score between 0 and 108 can be given [34].

### 2.7. Bioinformatic Analysis

The 32-bit version of USEARCH [35] and mothur [36] was used in combination with several in-house programs for bioinformatics analysis of the sequence data.

Following tag identification and trimming, all sequences from all samples were pooled. Paired-end reads were merged, truncating reads at a quality score of 4, requiring a merged read length between 300 and 600 bp in2 length. Sequences with ambiguous bases, without a perfect match to the primers, a homopolymer length greater than 8, or more than one expected error, were discarded and primer sequences trimmed. Sequences were strictly dereplicated, discarding clusters smaller than 5.

Sequences were clustered at 97% sequence similarity, using the most abundant strictly dereplicated reads as centroids and discarding suspected chimeras based on internal comparison. Additional suspected chimeric OTUs are discarded based on comparison with the Ribosomal Database Project classifier training set v9 [37] using UCHIME [38]. Taxonomic assignment of OTUs is done using the method by Wang et al. [39] using the database Ribosomal Database Project. Downstream analysis was based on rarified data and performed in R version 3.3 (R Development Core Team (2008). R: A language and environment for statistical computing. R Foundation for Statistical Computing, Vienna, Austria. ISBN 3-900051-07-0, URL http://www.R-project.org.) using the vegan package (Jari Oksanen, F. Guillaume Blanchet, Michael Friendly, Roeland Kindt, Pierre Legendre, Dan McGlinn, Peter R. Minchin, R. B. O’Hara, Gavin L. Simpson, Peter Solymos, M. Henry H. Stevens, Eduard Szoecs and Helene Wagner (2016). vegan: Community Ecology Package. R package version 2.4-1. https://CRAN.R-project.org/package=vegan).

### 2.8. Statistical Analysis

All statistical analyses were conducted in SPSS version 25 (IBM, Armonk, NY, USA). T-test was used for interval scaled data. Chi-square tests were used to analyze nominal variables. When appropriate, Fisher’s exact test was used. All results are presented as mean ± standard deviation (SD). A *p*-value < 0.05 was considered to be statistically significant.

## 3. Results

### 3.1. Microbiome of PD Patients

In line with former studies, we could confirm a difference in bacterial composition between patients with PD and HC. None of the changes reached significance (Figure 2). We could show a tendency towards a reduction of *Prevotellaceae*, *Bacteroidetes* and the genus *Butyricimonas* and *Odoribacter* in PD (Figure 3). Moreover, we could show a relative increase of *Actinobacteria* and *Firmicutes* compared to healthy controls. Furthermore, *Negativicutes* and the phylum *Proteobacteria* showed a tendency to increase.

### 3.2. Microbiome Composition before and after Combined Treatment

Our data show that the UPDRS III improved and decreased significantly after therapy (*p*= 0.0004). An analysis of the subgroup also showed a significant decrease in the group with vegetarian diet (*p* = 0.033) and the group with additional bowel cleansing (*p* = 0.005) (Figure 4). Analysis of the subgroups revealed no change of the Shannon index before and after therapy in general (Figure 5). 

Interestingly, there was a significant association in the generalized linear model between the Shannon index and the UPDRS III in all groups with therapy (*p* = 0.006) (Figure 6). Furthermore, there was a correlation between the abundance of *Ruminococcaceae* and the UPDRS III (*p* = 0.0003) (Figure 7). Only the abundance of *Clostridiaceae* was significantly reduced after herbal enema (*p* = 0.043) (Appendix A).

### 3.3. Dosage of Levodopa before and after One Year

In a one year follow up of patients who received combined bowel cleansing and dietary intervention, a decrease in levodopa dose of 36 mg/day (mean) was noted. The intervention group with an isolated dietary intervention had an increase of 100 mg of daily levodopa after one year (0.173) (Table 4). 

## 4. Discussion

In our study, we could corroborate the former results on the dysbalance of the microbiome related to PD, that is, a relative increase of *Actinobacteria* and *Firmicutes* compared to healthy controls. Nevertheless, none of those changes reached significance (Figure 2). Additionally, a tendency towards a reduction of *Prevotellaceae*, *Bacteroidetes* and the genus *Butyricimonas* and *Odoribacter* in PD was present (Figure 3). Furthermore, *Negativicutes* and the phylum *Proteobacteria* showed a tendency to increase.

In both subgroups – combined dietary intervention plus bowel cleansing and the isolated dietary intervention group - we observed a significant clinical improvement as quantified by UPDRS III. Furthermore, the study revealed a significant association in the generalized linear model within the pooled group between the Shannon-Index and the UPDRS III (*p* = 0.006), an indicator of bacterial diversity in the gut of PD patients. Interestingly, there was a positive correlation between the UDPRS III and the frequency of *Ruminococcaceae*. Additionally, the abundance of *Clostridiaceae* was significantly reduced after enema.

Recently there have been various studies with different approaches examine the gut microbiome in patients with PD. It is now agreed that a shift of the microbiome, that is, towards dysbiosis, is likely linked to disease pathogenesis and not to single aberrant bacterial strains. The complex changes of the microbiome composition might affect metabolic, immunological and homeostatic functions of the host organism [40,41]. In this context, Keshavarzian et al. showed that in PD a pro-inflammatory dysbiosis is prevalent and as a result, alpha-synuclein is increased in the gut of PD patients. Therefore, they found reduced levels of “anti-inflammatory” butyrate-producing bacteria from the genus *Blautia*, *Coprococcus* and *Roseburia* [42,43]. Furthermore, an increase of “pro-inflammatory” Proteobacteria of the genus *Ralstonia* was documented [8]. In our cohort we could also confirm a tendential increase of *Proteobacteria* and *Firmicutes* [4,5,6] in the gut of PD patients as a sign of dysbiosis, a potential ‘pro-inflammatory’ state and in case of *Firmicutes* a potential negative influence on the intestinal wall integrity [44,45]. In addition, a negative correlation between the frequency of *Firmicutes* and the duration of PD has been previously detected, meaning a reduced frequency in elder patients [8]. In line with several studies we could confirm a tendency towards the reduction of the *Prevotellaceae* [7] and *Bacteroidetes* [5] in PD. Furthermore, in this study we found a significant correlation between the frequency of *Ruminococcaceae* and the UPDRS-III (Figure 8). This confirms the relevance of the microbiome for the clinical course. Along with two former studies, we found a reduced but not significant frequency of the genus *Prevotella* [5]. Moreover, *Prevotella* enrichment has been linked to a rich-fiber diet, which is the primary source for SFCA including butyrate [46,47]. A lack of Butyrate has been substantiated in PD and is associated to disrupt barrier function and enhanced promote inflammation [7,46,48]. In this context Unger et al. detected a decreased level of SCFA within stool samples of PD patients [7]. Additionally, we could corroborate an often-observed relative increase of *Actinobacteria* compared to healthy controls [4,5,7]. Many species of the phylum can produce folate [49]. According to this, an often discussed folate deficiency that can have major adverse effects on the developing human nervous system could not be confirmed in PD [50,51]. Furthermore, also in our cohort, the frequency of the genus *Butyricimonas* and *Odoribacter* were tendentially reduced [5]. Both bacteria are producers of short-fatty acids [52,53]. In summary our data support alterations of the microbiome in PD. Therefore, we have tested the influence of a therapy on the microbiome. 

A well controllable approach to positively influence the microbiome is a vegetarian diet during an inpatient stay. A vegetarian diet appears to be beneficial to human health by promoting the development of a more diverse microbiome [54,55]. Therefore, a positive association has been found between the alpha-diversity and long-term fruit and vegetable intake and Martinez et al. detected an increase of microbial diversity by adding whole-grain barley and brown rice to the diet [56,57]. Furthermore, a pro-inflammatory microbiome has been found in obese participants, with a reduction in the Bacteriodetes:Firmicutes ratio and an increase in Proteobacteria by Verdam et al. [58]. Additionally, non-digestible carbohydrates not only act as prebiotics by promoting the growth of beneficial microorganisms but also reduce proinflammatory cytokine production in humans [47,59]. A metabolic product of some intestinal bacteria is SCFA, which is fermented by the families *Prevotellaceae* and *Lachnospiraceae* and the genera *Akkermansia*, *Blautia*, *Roseburia* and *Faecalibacterium* from a fiber-rich diet. All SCFA producers are decreased in PD except the genus *Akkermansia* [7,42]. Hence, their decrease in PD may be associated with reduced SCFA levels. SCFA are essential for intestinal barrier function, regulation of intestinal motility and immunological processes in the body [60]. They also lead to a down-regulation of pro-inflammatory cytokines (IL-8, IL-6, IL-1β, IFN-γ and TNF-α) and the promotion of colonic regulatory T cell differentiation [61,62]. Furthermore, the propionate receptor FFAR3 was discovered on human brain endothelium and an influence of propionate on the BBB via a CD-14-dependent mechanism has been detected [63,64]. In this regard, 500–500 mmol of SCFA are produced in the gut daily, 95% of SCFAs produced are absorbed within the colon and a level of 17 pmol/mg butyrate and 18,8 pmol/mg propionate within the brain tissue have been reported [65]. It has also been shown that oral administration of SCFA promotes the maturation of microglial cells, which are essential for the maintenance of tissue homeostasis within the brain [66]. Additionally, an influence on gene expression, an endocrine (GLP-1, leptin, ghrelin and insulin), a vagal and humoral pathway as well as many transporters and receptors with partly high affinity to the SCFA have been explored [65,67]. A substitution of SCFA might, in addition to already proven neuroprotective effects, also positively influence the course of the disease via its anti-inflammatory action [43,48]. For this reason, we supplemented Ghee including butyrate and propionate to the vegetarian diet [68].

Rectal enemas are a treatment option for patients with IBD (usually with a drug supplement) and another option for influencing the microbiome. Four studies have already shown significant changes in composition and diversity under enemas [69,70,71,72]. Additionally, one study has observed changes in intestinal metabolites. Therefore, significant changes including an increase or decrease of 32 metabolites were documented immediately after enema. Most of these metabolites were amino acids [73]. 

For the first time, we investigated in this proof-of-concept study the influence of an intervention including dietary intervention containing SCFA and enema in a small cohort of patients with PD on the microbiome and related it to the clinical course. Different influences of the microbiome on intestine-associated processes are being considered in the current scientific debate. Increased amounts of pro-inflammatory bacterial species have been determined. As already discussed, influence on inflammatory processes within the intestinal wall and thus indirect influence permeability and might lead potentially to a-synuclein secretion within the intestinal wall. In this respect, in addition to the increase of SCFA through a vegetarian diet, anti-inflammatory effects could be described through a vegetarian diet [10].

Surprisingly, there was no difference in beta-diversity and no significant differences before and after intervention in the relative frequency (Figure 5 and Figure 6). This might be due to the small sample size.

Interestingly, we could detect a negative significant association in the generalized linear model between the Shannon index and the UPDRS III in the combined data of the interventional groups (*p* = 0.006) (Figure 9). Nevertheless, it must be mentioned that besides a strong not significant, negative correlation in the enema group, there was no tendency for correlation in the nutrition group. This could be due to the short intervention period. For example, the influence of enema on the microbiome is already apparent after a few days, whereas the influence of vegetarian diets on the microbiome probably takes a longer [73]. The Shannon index reflects microbial diversity. Therefore, a high microbial diversity or high Shannon index is considered to be healthy [74]. In addition, the UPDRS improved after treatment (Figure 4). Contrary to the stable to increasing levodopa demand as disease progression documented in the literature [75,76], we were able to report a significant reduction of the dopa equivalence dose one year after intervention with enema (Table 4). Nevertheless, an increase of the Shannon-Index before and after therapy could not be documented (Figure 7). 

Furthermore, there was a positive correlation between the abundance of *Ruminococcaceae* and the UPDRS III (*p* = 0.0003) (Figure 9). Already, two previous studies have found an increased amount of the species in the intestine in patients with PD [4]. A research group including Hill-Burns found an increase of the amount of *Ruminococcaceae* associated with disease duration [4]. Considering that the UPDRS III increases during the course of the disease, this strain may be important for possible therapeutic interventions. The abundance of *Clostridiaceae* was significantly reduced after herbal enema (*p* = 0.043) (Appendix A). *Clostridiaceae* are a family of the class *Clostridia.* Further families of this bacterial class are putative proinflammatory bacteria and were documented to be increased in PD [8]. In an animal study *Clostridiaceae* were more present in dogs with a meat diet and the abundance correlated negatively with the levels of SCFA [77]. According to Kesharvzian et al., the reduction of *Clostridiaceae* might reduce inflammation within the colon and this might reduce alpha-synuclein secretion [8]. At the clinical level, our data show that the UPDRS III improved and decreased significantly after therapy (*p* = 0.0004). An analysis of the subgroup also showed a significant decrease in the group with vegetarian diet (*p* = 0.033) and the group with additional bowel cleansing (*p* = 0.005). However, it must be mentioned that the vegetarian diet group consisted of only 6 patients and has a high standard deviation (Figure 4). The detailed data of the individual changes of the UPDRS III can be seen in Appendix A. According to the recently documented influence of *Enterococcus faecalis* via tyrosine decarboxylase and *Eggerthela lenta* via dopamine dehydroxylase on the metabolism of levodopa we could not find any impact on the relative abundance after intervention (Appendix A). Furthermore, no correlation between the levodopa dose and the relative abundance could be detected (Appendix A). According to the presented data a limiting factor in this study is the significant difference in age, since age is considered to be a major influencing factor on the microbiome along with gender [44]. 

In summary, there is growing evidence that changes in the gut microbiome and metabolome may have a direct or indirect impact of the healthy and diseased brain, as is the case for PD. Interventions that may skew the gut microbiome composition and may have a therapeutic effect on the course of the disease. Although small in size, our proof-of-concept study suggests that bowel cleansing, a well-established treatment and good tolerable intervention, in combination with a dietary intervention including SCFA have a positive effective not only on the gut microbiome but may also have a beneficial effect on the clinical course in PD. Nevertheless, as mentioned above the potential side effects including orthostatic dysregulation, electrolyte imbalance, exsiccosis, infections, abdominal pain, diarrhea, nausea and vomiting must be considered.

## 5. Limitations

One limitation of our study is the significant age difference of the two groups, because previous studies have shown age and sex related changes in the microbiome [44,78]. Furthermore, it is also important to mention that the intervention group is too small to make definitive statements with a case number of 16 participants and that moderate changes of the microbiome can develop over disease stage.

## 6. Conclusions

As already proven in previous studies, dysbiosis is present in patients with PD. Additionally, a link between the microbiome and the clinical course seems likely. Dietary intervention and bowel cleansing are sufficient methods to impact the gut microbiome in patients with PD. Therefore, a positive impact on the clinical course is feasible. Nevertheless, studies with a greater case number are needed for sufficient conclusions.

## Figures and Tables

**Figure 1 cells-09-00376-f001:**
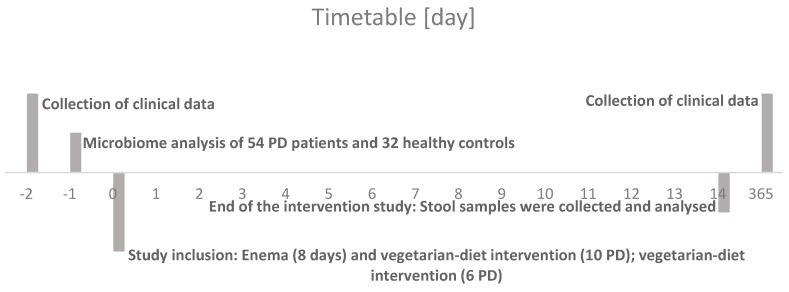
Timetable of study characteristics in days.

**Figure 2 cells-09-00376-f002:**
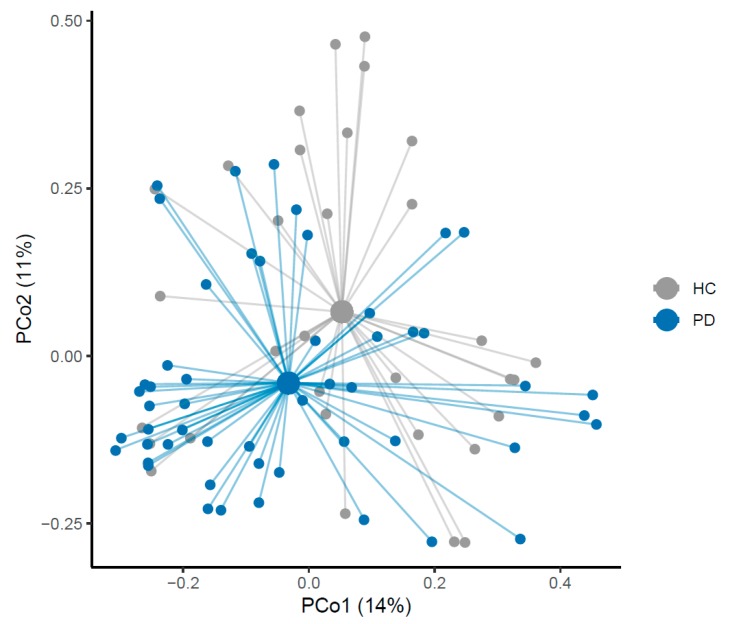
Principal Coordinates Analysis (PCoA) of unweighted UniFrac. HC: Healthy Controls; PD: Parkinson Disease.

**Figure 3 cells-09-00376-f003:**
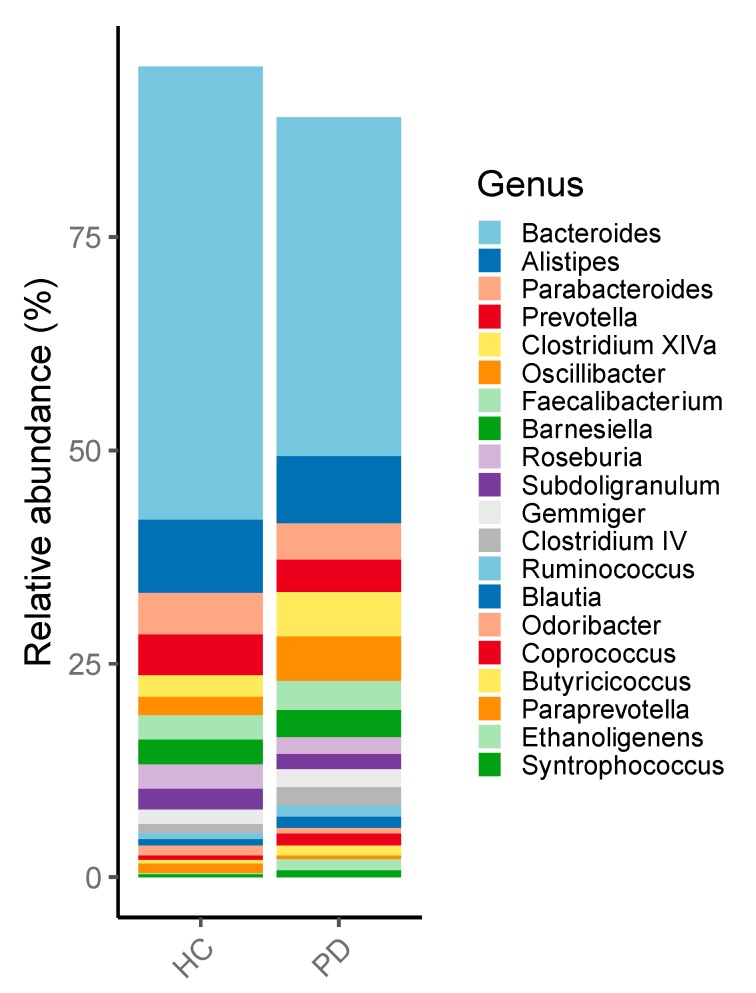
Relative abundance of the genus level in patients with Parkinson disease and healthy controls. HC: Healthy Controls; PD: Parkinson Disease.

**Figure 4 cells-09-00376-f004:**
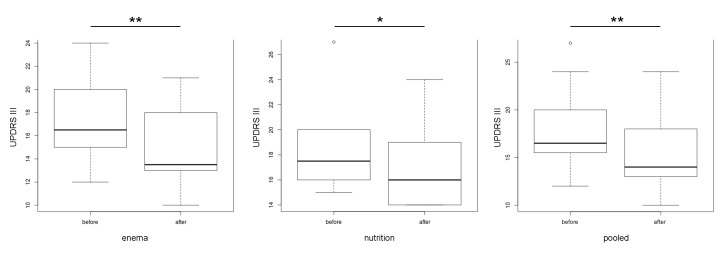
Unified Parkinson Disease Ratings Scale (UPDRS) III before and after one year after the 14 days interval of treatment for each individual group (enema, only nutrition) and for both groups together (pooled) (* *p* < 0.05; ** *p* < 0.01).

**Figure 5 cells-09-00376-f005:**
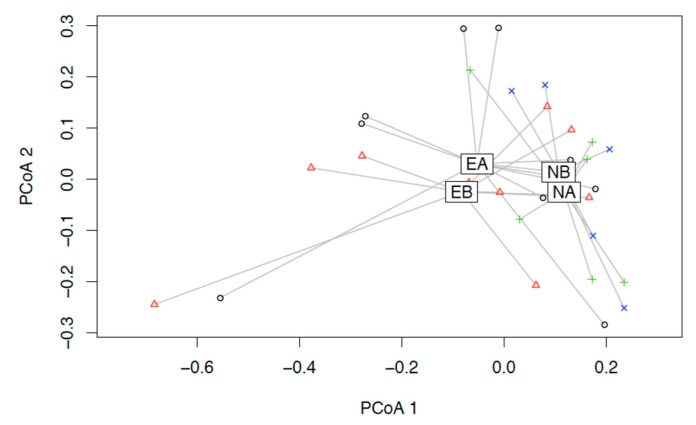
Principal Coordinates Analysis (PCoA) of Bray-Curtis- dissimilarity for the beta-diversity before and after intervention. EB: enema before, EA: enema after intervention; NB: nutrition before, NA: nutrition after intervention.

**Figure 6 cells-09-00376-f006:**
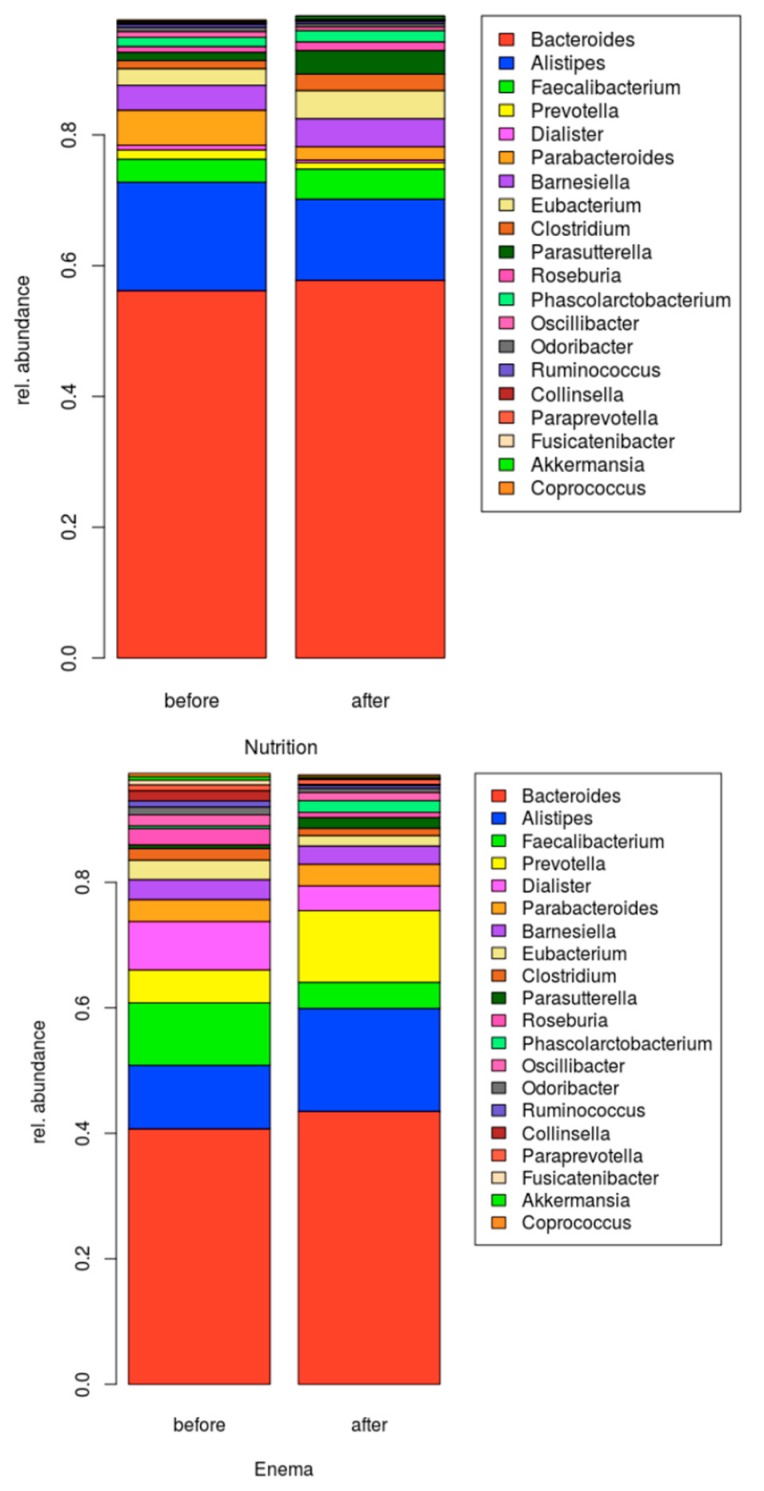
Relative abundance of the genus level in patients with Parkinson disease before and after intervention.

**Figure 7 cells-09-00376-f007:**
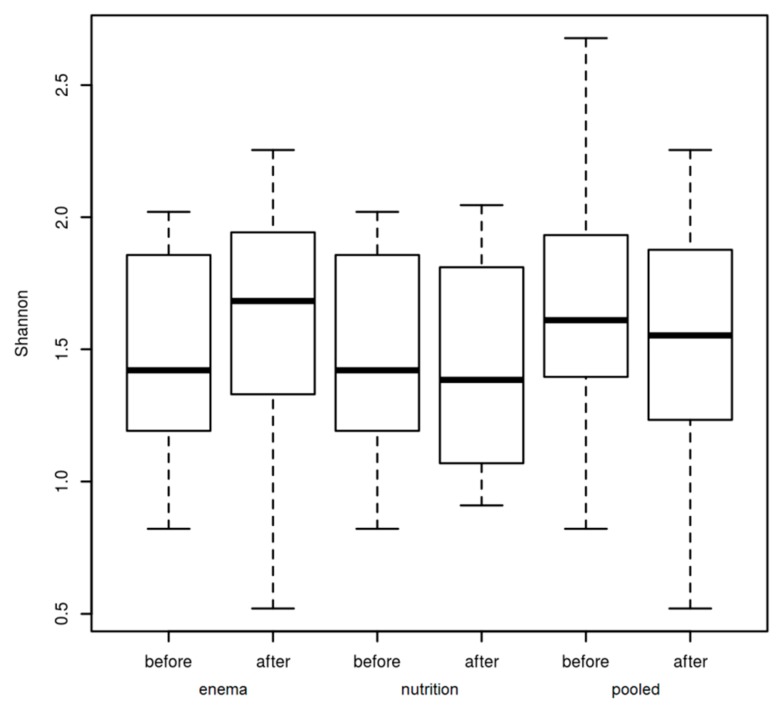
Shannon index before and after the 14 days of therapy for each individual group (enema, only nutrition) and for both groups together (pooled). Box-Whisker-Plots (minimum, 25%-quartile, median, 75%-quartile, maximum).

**Figure 8 cells-09-00376-f008:**
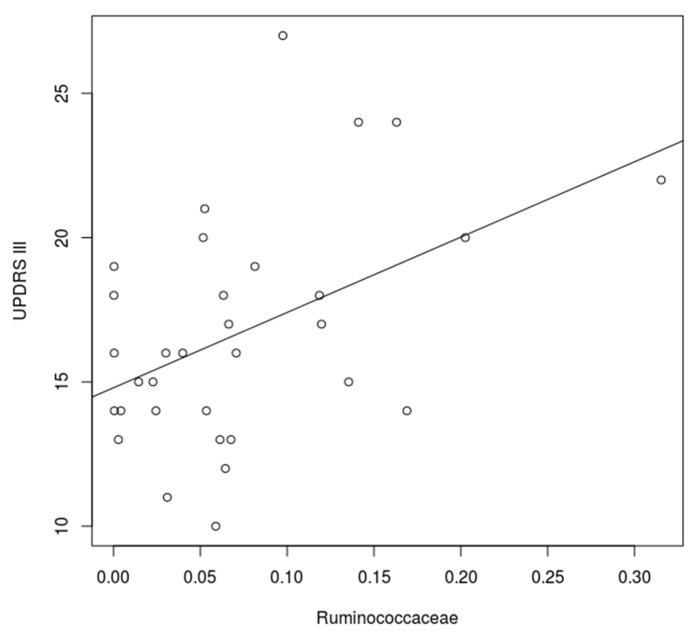
Correlation between frequency of *Ruminococcaceae* and UPDRS III. There was a significant correlation between the frequency of *Ruminococcaceae* and UPDRS III in both interventional groups combined (*p* = 0.003).

**Figure 9 cells-09-00376-f009:**
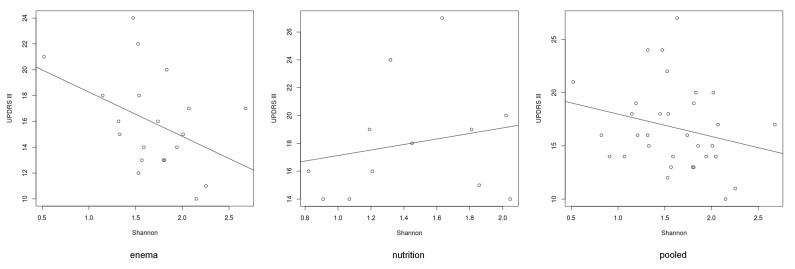
Correlation between the Shannon-Index and the UPDRS III. There was no significant correlation between the Shannon-Index and the UPDRS III but a significant association in the generalized linear model within the pooled group (*p* = 0.006).

**Table 1 cells-09-00376-t001:** Demographic data and characteristics including standard deviation or percentage data of patients without therapeutic intervention and healthy subjects (the given data are complete for all participants).

Characteristics	PD (*n* = 54)	HS (*n* = 34)	*p*
Female sex, n (%)	27 (50%)	20 (58.8%)	0.425 ^a^
Age, years, mean ± SD	61 (±9.2)	52.8 (±12.6)	0.00 ^b^
Disease duration, years, mean ± SD	9.1 (±5.8)		
BMI	26.15 (±4.5)	26.1 (±5.5)	0.952 ^b^
Subgroupsakinetic-rigid	23 (42.6%)		
equivalent	25 (46.3%)		
tremordominant	6 (11.1%)		
Medication			
L-Dopa (mg) daily dose	388 (±276)		
Benserazid	37 (68.5%)		
Carbidopa	20 (37%)		
Entacapon	13 (24.1%)		
MAO-B Hemmer	39 (72%)		
Dopamine agonists	40 (74.1%)		
Amantadine	25 (46.3%)		
Anticholinergics	1 (1.9%)		
Proton pump inhibitor	5 (9.3%)	2 (5.9%)	0.622 ^c^
Vegetarian	5 (9.3%)	3 (8.8%)	0.333 ^c^
Mostly (rarely meat)	16 (29.6%)	5 (14.7%)	0.836 ^c^
Vegan	1 (1.9%)	0 (0%)	0.433 ^c^
How often meat per week			0.069 ^c^
Non	9 (16.7%)	4 (11.8%)	
1–2	14 (25.9%)	5 (14.7%)	
3–5	20 (14.8%)	9 (26.5%)	
6–7	8 (37%)	13 (38.2%)	
Bristol Stool scale			0.418 ^c^
1	21 (38.9%)	9 (26.5%)	
2	13 (24.1%)	12 (35.3%)	
3	12 (22.2%)	12 (35.3%)	
4	8 (14.8%)	5 (14.7%)	
5	0	0	
6	0	0	
7	0	0	
Unified Parkinson Disease Rating Scale (UPDRS)			
UPDRS I	2 (± 1.9)		
UPDRS II	7.2 (± 4.6)		
UPDRS III	14.9 (± 10.4)		
UPDRS IV	1.8 (± 2.3)		
UPDRS V	1.8 (± 0.6)		
UPDRS VI	0.9 (± 0.1)		

^a^ Fisher’s exact test; ^b^
*t*-test; ^c^ X2-Test.

**Table 2 cells-09-00376-t002:** Demographic data and characteristics of patients with therapeutic intervention (the given data are complete for all participants).

Characteristics	PD (*n* = 16)
Female sex, n (%)	10 (63%)
Age, years, mean ± SD	64 ± 5.4
Disease duration, years, mean ± SD	8.6 ± 4.1
BMI	26.7 ± 4
Vegetarian	1 (6.3%)
Mostly (rarely meat)	4 (25%)
Vegan	1 (6.3%)
How often meat per week	
Non	0 (0%)
1–2	5 (31.3%)
3–5	8 (50%)
6–7	3 (18.8%)

**Table 3 cells-09-00376-t003:** Ingredients during the two-week diet which all patients with Parkinson’s disease (PD) have received.

Nutrition’s	
Ghee	Ghee is a pure clarified fat exclusively obtained from milk, cream or butter. Almost a total removal of water and non-fat solids with a total fat content of 62% is achieved.
Vegetables	Onions, garlic, potatoes, carrots, chives, spinach, lentils, tomatoes, auberges, ginger, zucchini, rucola, cauliflower, fennel, broccoli, celery, leek, chicory, swiss chard, Chinese cabbage, kohlrabi, Muscat pumpkin, beetroot, chickpeas, mung beans, Lollo rosso, spring onions, cucumber, iceberg lecture, paprika.
Fruits	Lemon, raspberry, strawberry, raisin, apple, radish, mango, figs, coconut, physialis, pineapple, peach, plums, oranges, grapes, cantaloupe melon, pomegranate, kiwi, banana, grapefruit, avocado, olives.
Cereals	Spelt, wheat, rice (basmati), rye, oats, millet, barley, semolina, maize.
Milk and egg products	Whole milk, eggs, quark, cream cheese, yogurt, sour cream, rice pudding, low-fat curd cheese.
Other	Noodle (spaghetti, penne), amaranth, bulgur, quinoa, ascorbic acid.
Spices and herbs	Vanilla, sugar, salt, pepper, coriander, cinnamon, chili, cardamom, cane sugar, basil, mint leaves, oregano, mustard, rosemary, marjoram, parsley, curry leaves, thymus, saffron, bay leaves, tridosha curry, lovage, star anise, ajwain, chervil, lime leaves.
Nuts, seeds and kernels	Sesame, almonds, pumpkin seed, cashew, kernel, fennel seed, nutmeg, sunflower seeds, hazelnuts.
Oil and vinegar	Olive oil, pumpkin seed oil, balsamic vinegar, walnut oil, sesame oil.
Other	Rose water, honey, maple syrup, soy sauce.

**Table 4 cells-09-00376-t004:** Levodopa-equivalent daily dose before and 1 years after therapy including dietary intervention with or without enema (The given data are complete for all participants).

	PD without Enema (*n* = 6)	PD with Enema (*n* = 10)	*p*-Value
Cum. dopamine doses before therapy (mg)	263.3 (147 mg)	537.6 (440.4 mg)	0.388 ^a^
Cum. dopamine doses one year after therapy (mg)	346.7 (156.9 mg)	481.4 (459.8 mg)	0.314 ^a^
Difference	83.3 mg (182.8 mg)	−56.1 mg (184.5 mg)	

^a^*t*-test.

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
