# Peer review of "Interventional Influence of the Intestinal Microbiome Through Dietary Intervention and Bowel Cleansing Might Improve Motor Symptoms in Parkinson’s Disease"

_cells, 2020, doi:10.3390/cells9020376_

Round 1
Reviewer 1 Report
Hegelmaier and colleagues present a small clinical study aimed at demonstrating that the intestinal microbiome is altered in PD patients and dietary/enema interventions are beneficial on clinical parameters of PD. The authors themselves state that critical limitations of the study are the age differences and the small number of participants. While the latter could not be addressed as a consequence of patient participation, the former is a serious issue as the authors are studying idiopathic (late onset) PD. Comparison with 18 year-olds is thus unsuitable for the purpose.
In addition, the study has several issues and missing results which hamper conclusive statements, conclusions are not supported and results are overinterpreted. Here are a few examples:
In 3.1 the authors state they confirm differences in bacterial composition in PD, but the following sentence reads "non of the changes reached significance". This is an overinterpretation; Fig 5: the association is significant but the direction is the opposite in the 3 groups. This is a critical difference that finds no mention whatsoever. In addition, the correlation is not significant, which has also been "hidden" in the discussion/interpretation; In 3.3, the authors compare the L-dopa intake at 1 year from intervention. These data are almost useless if not compared to PD patients with no intervention; Fig 3: the UPDRS III in the "nutrition" group is extremely variable, this needs addressing. Again, PD patients with no intervention at the same timepoints should be included; The authors show a correlation of UPDRS III only with Ruminococcaceae. What about the gena with stronger trends? For a proper evaluation of the effects of the interventions, the authors should include healthy controls that underwent the same treatments and analyze their microbiome before and after; Why do the authors only report levels of Clostridiaceae after treatment? What about the whole microbiome gena? In the discussion, the authors do not mention that the (modest) changes they observe cannot be ascribed to causality as observed in over disease states. This is a critical discussion point; Fig. 4 indicates that the treatments have no effects on the microbiome, but the authors state several times that they doAuthor Response
Many thanks for the detailed criticism. In the discussion we emphasized that none of the results were significant, but only showed tendencies (line 315, 394-398). Regarding Figure 5 (now Figure 6), we have highlighted and discussed the large variations in the correlations between the different groups in line 369-273.
Of course, a comparison group with patients with PD without intervention would have been interesting as a comparison. Unfortunately, we did not record this in this study, but we would have assumed a rather stable dopamine equivalent dose and a stable UPDRS III within one year.
According to figure 3 (now 4) we have addressed your concerns in line 416-417.
Apart from the correlation between Ruminoccocaeand UPDRS III, the analyses did not reveal any further correlations between the clinical data and the gena.
There are already several studies that have looked at the influence of enema and vegetarian diets on the microbiome in healthy volunteers (Nagata N, 2019; Pagliai G, 2019). We have taken this as a prerequisite for this study.
Regarding the Shannon index, the therapy had no influence, but despite the small sample size, significant changes in Clostridiaceaewere observed after therapy. Furthermore, an influence of therapies on the microbiome in healthy patients has already been shown. Additionally, we added your concerns, regarding the changes of the microbiome over time within the disease to the limitations at the end of the manuscript in line 446-447.
Reviewer 2 Report
In this interesting manuscript, Hegelmaier et al describe a cohort of persons with PD provided a vegetarian diet and/or an enema, and measure changes to both the microbiome and PD symptoms. While on its face this is a unique and interesting study, this reviewer has a number of concerns that should be addressed prior to publication. In particular, these revolve around aspects of study design, interpretation, and data display.
Major Concern A. The authors demonstrate that following nutritional therapy or enema intervention, persons with PD have decreased UPDRSIII scores (Fig. 3). While a striking and important result, there are a few significant caveats that must be addressed here.
A1. It is not clear from reading the methods or results the precise timeline of these interventions and the subsequent analysis. How long after nutrition or enema was the UPDRS observations performed? When were the subsequent microbiome samples collected? This reviewer would suggest a graphical timeline representation of when samples were collected, when and how long nutritional and enema interventions took place, when UPDRS testing occurred, etc. This is necessary for proper interpretation of the results by the reader (and by this reviewer!).
A2. A major caveat of this study is the lack of an untreated control group. Over the timeline tested (which as mentioned was unclear to this reviewer), how would a person with PD (but no nutrition or enema intervention) change in UPDRS or other parameters?
A3. In Figure 3, arguably one of the most important figures in the manuscript, the authors present compiled data. Here, it would be extremely beneficial for the authors to display individual points representing each patient and how each one responded (or didn't respond) to the indicated treatments. As the numbers are small (16 for instance) this could even be provided in a supplemental table. This is also where an untreated control group would be extremely important.
A4. The authors display UPDRSIII scores following treatments, but did other UPDRS parameters change? Did changes to the Bristol Stool scale (as measured at baseline) occur following treatment?
Major Concern B. The authors indicate taxonomic differences in the microbiome between PD and control, that while not significant, generally agree with the established literature. However, the analysis of the microbiome following the various treatments is extremely limited and is unclear.
B1. The authors state that after the therapy, there is no difference in microbiome composition. However, the only evidence for this as presented is the Shannon Index (a measure of alpha diversity or richness) in Figure 4. It is entirely possible (in fact likely) that richness may not change, but that treatments resulted in changes to abundances of particular taxa (beta diversity) which could be measured with Bray-Curtis and UniFrac.
B2. Similarly, a major gap is the subsequent analysis of the microbiome in PD following treatment. Not only examining alpha and beta diversity, but graphical representations of any taxonomic changes and clustering of populations by treatment. Recapitulating Figure 2 and 3 for pre- vs. post-treatment microbiomes would be necessary for proper interpretation and support of the authors' claim that changing the microbiome is beneficial for PD.
B3. In Figure 5, the authors demonstrate mild correlations between alpha diversity and UPDRSIII scores. This reviewer assumes this is only post treatment, but such correlations prior to treatment would be beneficial and help to support the argument that changes to such diversity are actually correlated to disease symptoms.
Minor concerns, generally presented in order of appearance-
While the Parkinson's Disease Questionnaire is cited in the methods, it is unclear what the the catagories "Normal" through "Heavy" actually mean. Similarly, UPDRS is not described in the methods. The authors state that ghee contains 30% butyrate. Was butyrate measured in their ghee preparations? No citation is given for butyrate content in ghee. The authors state that their diet is more increased in SCFA than others, but no citation is given for how much SCFA is present in the average vegetarian diet, etc. As mentioned in A1, the timing of the observations in the figures is often unclear. In addition to a graphical timeline, figure legends could be expanded to include more specific information regarding when samples were collected. For instance, is Fig. 6 after treatments? Which one? The authors make a surprising observation that enema reduced L-DOPA usage at 1 year post-treatment (Was this the same timing as the UPDRS follow ups? See point A1). Given the cited literature regarding L-DOPA metabolizers present in the gut microbiome, it should be possible for the authors to identify those specific taxa in their sequencing analysis and observe if there are any differences in Enterococcus and/or Eggerthella. (See Rekdal et al. Science 2019 and van Kessel et al. Nat. Comm. 2019). The authors should consider inclusion of supplemental tables to highlight the notable taxonomic changes in their 16s RNA sequencing analysis. Figure legends could be revised to include more pertinent information- such as timing, number of individuals per group, description of error bars, statistical analysis used, etc. Confidence intervals would also be useful to be displayed in the correlation analysis. It would be beneficial for the field if the sequencing data and associated metadata are made freely available in a public database such as the ENA or QIITA.
Author Response
Reviewer 2:
In this interesting manuscript, Hegelmaier et al describe a cohort of persons with PD provided a vegetarian diet and/or an enema, and measure changes to both the microbiome and PD symptoms. While on its face this is a unique and interesting study, this reviewer has a number of concerns that should be addressed prior to publication. In particular, these revolve around aspects of study design, interpretation, and data display.
Major Concern A. The authors demonstrate that following nutritional therapy or enema intervention, persons with PD have decreased UPDRSIII scores (Fig. 3). While a striking and important result, there are a few significant caveats that must be addressed here.
A1. It is not clear from reading the methods or results the precise timeline of these interventions and the subsequent analysis. How long after nutrition or enema was the UPDRS observations performed? When were the subsequent microbiome samples collected? This reviewer would suggest a graphical timeline representation of when samples were collected, when and how long nutritional and enema interventions took place, when UPDRS testing occurred, etc. This is necessary for proper interpretation of the results by the reader (and by this reviewer!).
Dear reviewer many thanks for the detailed criticism and for this advise. We have added a graphical representation of the timetable in the methodology section in line 175 (Figure 1).
A2. A major caveat of this study is the lack of an untreated control group. Over the timeline tested (which as mentioned was unclear to this reviewer), how would a person with PD (but no nutrition or enema intervention) change in UPDRS or other parameters?
We understand your concerns, unfortunately we do not have an untreated control group with Parkinson syndrome in this study. We have assumed a stable result of the levodopa equivalent units and the UPDRS within an untreated group.
A3. In Figure 3, arguably one of the most important figures in the manuscript, the authors present compiled data. Here, it would be extremely beneficial for the authors to display individual points representing each patient and how each one responded (or didn't respond) to the indicated treatments. As the numbers are small (16 for instance) this could even be provided in a supplemental table. This is also where an untreated control group would be extremely important.
Many thanks for the suggestion. We have added a corresponding graphic in the supplements under Figure S2 and cited this figure in line 418.
A4. The authors display UPDRSIII scores following treatments, but did other UPDRS parameters change? Did changes to the Bristol Stool scale (as measured at baseline) occur following treatment?
We agree with the reviewer on this aspect, but unfortunately, we have only documented UPDRS III in the course.
Major Concern B. The authors indicate taxonomic differences in the microbiome between PD and control, that while not significant, generally agree with the established literature. However, the analysis of the microbiome following the various treatments is extremely limited and is unclear.
B1. The authors state that after the therapy, there is no difference in microbiome composition. However, the only evidence for this as presented is the Shannon Index (a measure of alpha diversity or richness) in Figure 4. It is entirely possible (in fact likely) that richness may not change, but that treatments resulted in changes to abundances of particular taxa (beta diversity) which could be measured with Bray-Curtis and UniFrac.
Besides the Shannon index we looked at the frequency of the individual bacteria. Here we could only observe a decrease of Clostridiaceaeafter therapy (Figure S1).
B2. Similarly, a major gap is the subsequent analysis of the microbiome in PD following treatment. Not only examining alpha and beta diversity, but graphical representations of any taxonomic changes and clustering of populations by treatment. Recapitulating Figure 2 and 3 for pre- vs. post-treatment microbiomes would be necessary for proper interpretation and support of the authors' claim that changing the microbiome is beneficial for PD.
We agree with the reviewer. Figure 3 (now 4) shows the values before and after therapy. We do not have a representation of the relative abundance before and after the therapy. However, we have compared the frequencies of the individual bacteria before and after therapy and only observed a decrease in Clostridiaceae.
B3. In Figure 5, the authors demonstrate mild correlations between alpha diversity and UPDRSIII scores. This reviewer assumes this is only post treatment, but such correlations prior to treatment would be beneficial and help to support the argument that changes to such diversity are actually correlated to disease symptoms.
Figure 5 (now 6) is a representation of all measuring points. I.e. all 32 measurements before and after the therapy were displayed.
Minor concerns, generally presented in order of appearance-
While the Parkinson's Disease Questionnaire is cited in the methods, it is unclear what the the catagories "Normal" through "Heavy" actually mean. Similarly, UPDRS is not described in the methods. The authors state that ghee contains 30% butyrate. Was butyrate measured in their ghee preparations? No citation is given for butyrate content in ghee. The authors state that their diet is more increased in SCFA than others, but no citation is given for how much SCFA is present in the average vegetarian diet, etc. As mentioned in A1, the timing of the observations in the figures is often unclear. In addition to a graphical timeline, figure legends could be expanded to include more specific information regarding when samples were collected. For instance, is Fig. 6 after treatments? Which one? The authors make a surprising observation that enema reduced L-DOPA usage at 1 year post-treatment (Was this the same timing as the UPDRS follow ups? See point A1). Given the cited literature regarding L-DOPA metabolizers present in the gut microbiome, it should be possible for the authors to identify those specific taxa in their sequencing analysis and observe if there are any differences in Enterococcus and/or Eggerthella. (See Rekdal et al. Science 2019 and van Kessel et al. Nat. Comm. 2019). The authors should consider inclusion of supplemental tables to highlight the notable taxonomic changes in their 16s RNA sequencing analysis. Figure legends could be revised to include more pertinent information- such as timing, number of individuals per group, description of error bars, statistical analysis used, etc. Confidence intervals would also be useful to be displayed in the correlation analysis. It would be beneficial for the field if the sequencing data and associated metadata are made freely available in a public database such as the ENA or QIITA.
The classification of PDQ-39 is not standardized. For this reason we have removed the questionnaire from the evaluation.
We have added a paragraph in the methodology to explain the UPDRS in line 205-220.
We have added a source on the content of SCFA in ghee in line 372. Unfortunately, we have no source for the percentage of SCFA.
Short-term fatty acids (SCFA) are ingested through a diet rich in fibre. Thus bacteria ferment the fibre-rich food in the intestine to SCFA, which can then be absorbed.
We have added a source on the influence of vegetarian diets on SCFA in line 389-391 (Pagliai G, 2019). A graphical timeline is added in Figure 1 (line 175).
We have extended the figure legend with regard to the chronological classification (273-275; 278-280; 297; 299-301).
Furthermore, we have added a graphical representation of the points in time in line 175, as recommended above. It is shown here that the UPDRS III and L-dopa dose were recorded one year after the 14 days of therapy.
With regard to Enterococcus and/or Eggerthella, an individual determination was not made at the time of the analysis, as the publications were not yet available at that time.
Thank you very much for the reference regarding the database. We will add the sequencing data and associated metadata to the ENA database.
Reviewer 3 Report
The manuscript is in line with the currently popular trend in neuro-psychiatric medicine to look for the intervention that may alleviate symptoms of PD. Presented data basicly support the conclusions however there are few issues that I strongly encourage authors to address before the paper is ready to be published:
Abstract: Specify the intervention (dietary vs. enema – faecal?) within the methodology and results sections
Introduction: Gut microbiota does not contain 10 times more cells in comparison to human cells. It was re-evaluated in 2017
Materials &Methods:
HS should be age matched, as this is major contributor of microbiota composition. This should be explained within the discussion section Specify which microbiota metrics were calculated and what forResults:
Was the comparison of gut microbiota composition analysed before or after the intervention, if yes which one (diet, enema?) Were the subjects within these groups similar in their dietary habits? Diet is the main factor shaping microecological niche within the gut. This is critical for analysis in neuropsychiatric diseases as PD persons are more vunlerable to malnutrition Decribe axes if figures How much butyrate and propionate was ingested? Was the consumption of ghee controlled? By introducing prebiotics into our digestive tract we enhance their production thereby present study is unable to demonstrate whether the ingestion of substrates for SCFAs or SCFAs themselves play a major role in refaunizationDiscussion
Rephrase the frequency of f.i. Ruminococcaceae into bacterial counts (this is what OTU tabel contains) CIT: One study was able to demonstrate additional 330 changes in intestinal metabolites enema [66]. State what changes There is no data how do SCFAs work to improve PD symptoms There is no explanation about negatives of enemasAuthor Response
The manuscript is in line with the currently popular trend in neuro-psychiatric medicine to look for the intervention that may alleviate symptoms of PD. Presented data basicly support the conclusions however there are few issues that I strongly encourage authors to address before the paper is ready to be published:
Abstract: Specify the intervention (dietary vs. enema – faecal?) within the methodology and results sections.
We have added additional information on vegetarian diets and enema in lines 25-31 in the methodology and results of the abstract.
Introduction: Gut microbiota does not contain 10 times more cells in comparison to human cells. It was re-evaluated in 2017
We have considered the current data situation in line 61-62 and have cited the work of Sender et al. 2016.
Materials &Methods:
HS should be age matched, as this is major contributor of microbiota composition. This should be explained within the discussion section Specify which microbiota metrics were calculated and what for
We have mentioned the reviewer's concerns in the limitations in lines 439-440. Furthermore, we added this aspect within the discussion in line 414-416. In this study we calculated the Shannon index and unweighted Unifrac.
Shannon Index: A common metric in microbiome analysis to identify and compare
alterations in microbiome' diversity between HC and Parkinson group.
Unweighted Unifrac: UniFrac was used as a distance metric for comparing biological
communities. We used unweighted Unifrac for a qualitative analysis in
respect to the variation between individual microbioms.
Results:
Was the comparison of gut microbiota composition analysed before or after the intervention, if yes which one (diet, enema?) Were the subjects within these groups similar in their dietary habits? Diet is the main factor shaping microecological niche within the gut. This is critical for analysis in neuropsychiatric diseases as PD persons are more vunlerable to malnutrition Decribe axes if figures How much butyrate and propionate was ingested? Was the consumption of ghee controlled? By introducing prebiotics into our digestive tract we enhance their production thereby present study is unable to demonstrate whether the ingestion of substrates for SCFAs or SCFAs themselves play a major role in refaunization
The individual bacterial frequencies were analyzed before and after each intervention. Here only a decrease of Clostridiaceae was observed. During the inpatient stay all patients received the same diet for 14 days including 3g of ghee.
The dietary habits of the individual patients were the same during the study period. The dietary habits before the study period are listed in Table 1. Here, a slightly increased meat consumption in the PD group can be seen.
In this study we did not measure the concentration of SCFA, but it could already be shown in a preliminary study that a change in diet from a meat-based to a vegetarian diet increases the concentration of SCFA. In our study, a large proportion of the patients ate meat for inclusion in the study. For this reason, an increase in SCFA in the cohort can be assumed.
Discussion
Rephrase the frequency of f.i. Ruminococcaceae into bacterial counts (this is what OTU tabel contains) CIT: One study was able to demonstrate additional 330 changes in intestinal metabolites enema [66]. State what changes There is no data how do SCFAs work to improve PD symptoms There is no explanation about negatives of enemas
We have elaborated the results of the study in lines 336-338. As far as we know, there are no clinical data on the influence of SCFA on Parkinson's syndrome. Only the influence on non-clinical data, such as inflammation values, has been investigated so far. In addition, in the summary of the results we have again highlighted the side effect of enema (line 420-422).
Round 2
Reviewer 2 Report
This reviewer appreciates the author's response to the editorial concerns. While the author's have toned the manuscript in response to these critiques, the study still has some larger flaws (particularly the lack of a control group, or even comparison to historical 1 year PD follow-up data). The authors also do not perform any decent measure of microbiome structure before and after treatments (other than Shannon diversity). This is extremely important if the model is such that the treatment is acting through the microbiome.
In response to this reviewer’s concern that the microbiome be assessed by other metrics, the authors state that they performed Shannon index and also examined individual taxa. The method for assessing whether individual taxa was altered is unclear to this reviewer, by the authors description. It also fails to examine beta-diversity. Richness and evenness may not change (Shannon index), and no individual taxa may be significantly altered, but beta diversity (UniFrac, Bray-Curtis) may be changed as this is a measure of population structure as a whole. Small, non-significant differences in many taxa could still be indicative of a changed microbiome population.
The authors state that microbiome composition is now analyzed prior and after treatment, however it is not apparent to this reviewer. If the data exist to measure Shannon diversity before and after, a complete picture of the microbiome structure should be able to be performed as well. This is significantly lacking. If the treatment is acting on and through the microbiome as the author’s speculate, then changes to the overall structure of the microbiome, through beta diversity, PCA, etc. should be apparent.
Requires citation that dopamine equivalent and UPDRS would be unchanged after 1 year. This reviewer believes the literature indicates that UPDRS increases rather steadily each year.
While the manuscripts describing Enterococcus and Eggerthella metabolism of l-DOPA may not have been published at the time of this study, the authors should easily be able to examine their sequencing data for OTUs corresponding to these taxa and assess whether changes to these taxa correspond with their L-DOPA equivalent metric.
Author Response
In response to this reviewer’s concern that the microbiome be assessed by other metrics, the authors state that they performed Shannon index and also examined individual taxa. The method for assessing whether individual taxa was altered is unclear to this reviewer, by the authors description. It also fails to examine beta-diversity. Richness and evenness may not change (Shannon index), and no individual taxa may be significantly altered, but beta diversity (UniFrac, Bray-Curtis) may be changed as this is a measure of population structure as a whole. Small, non-significant differences in many taxa could still be indicative of a changed microbiome population.
We have added a figure in line 258 to the manuscript. We show the representation of the beta-diversity of both interventional groups before and after intervention. We could demonstrate no significant changes in beta-diversity between the groups in context of both interventions.
The authors state that microbiome composition is now analyzed prior and after treatment, however it is not apparent to this reviewer. If the data exist to measure Shannon diversity before and after, a complete picture of the microbiome structure should be able to be performed as well. This is significantly lacking. If the treatment is acting on and through the microbiome as the author’s speculate, then changes to the overall structure of the microbiome, through beta diversity, PCA, etc. should be apparent.
We agree with the reviewer that a representation of the microbial structure is interesting for further evaluation. For this reason, we reanalyzed our data to add a representative figure for bacterial composition for both treatment groups in line 263 (Figure 6). As mentioned in the previous paragraph we added a figure for beta-diversity in line 258.
Requires citation that dopamine equivalent and UPDRS would be unchanged after 1 year. This reviewer believes the literature indicates that UPDRS increases rather steadily each year.
Thank you for the additional information. In the literature we have found indications for a stable to increasing levodopa requirement. We have discussed the literature (Vorovenci, 2015; Nonnekes, 2016) in the context of table 4 in lines 396-398.
While the manuscripts describing Enterococcus and Eggerthella metabolism of l-DOPA may not have been published at the time of this study, the authors should easily be able to examine their sequencing data for OTUs corresponding to these taxa and assess whether changes to these taxa correspond with their L-DOPA equivalent metric.
We have additionally reanalyzed the relative abundances of Enterococcus and Eggerthela in respect to the individual treatment and cumulative levodopa dose. There was no significant difference between before and after the intervention (Figure S3). In addition, in Figure S4 we compared the relative frequency with the cumulative levodopa equivalent dose (Figure S4). No significant correlation was found here. We have discussed this in the manuscript in lines 424 - 428.
Reviewer 3 Report
I recommend to add the following articles in the discussion section (regarding SCFAs impact on CNS)
Hoyles et al., Microbiome. 2018 Mar 21;6(1):55
Erny et al., Immunology. 2017 Jan;150(1):7-15.
https://www.ncbi.nlm.nih.gov/pubmed/27346602
Dalile et al., Nat Rev Gastroenterol Hepatol. 2019 Aug;16(8):461-478
Author Response
Hoyles et al., Microbiome. 2018 Mar 21;6(1):55
Erny et al., Immunology. 2017 Jan;150(1):7-15.
https://www.ncbi.nlm.nih.gov/pubmed/27346602
Dalile et al., Nat Rev Gastroenterol Hepatol. 2019 Aug;16(8):461-478
Thank you for the supplementary literature, we have discussed it additionally in line 355-361.